# Superhuman performance on sepsis MIMIC-III data by distributional reinforcement learning

**Markus Böck[1], Julien Malle[1], Daniel Pasterk[1]\*, Hrvoje Kukina[1], Ramin Hasani[2], Clemens Heitzinger[1,3]**

1 Technische Universität Wien (TU Wien), Vienna, Austria, 2 Massachusetts Institute of Technology (MIT), Cambridge, MA, United States of America, 3 CAIML (Center for Artificial Intelligence and Machine Learning), TU Wien, Vienna, Austria

These authors contributed equally to this work.
\* daniel.pasterk@tuwien.ac.at

**Data Availability Statement:** The data underlying the results presented in the study are available from https://mimic.physionet.org/ These are these third party data, which are accessible in the same manner as the authors.

## Abstract

We present a novel setup for treating sepsis using distributional reinforcement learning (RL). Sepsis is a life-threatening medical emergency. Its treatment is considered to be a challenging high-stakes decision-making problem, which has to procedurally account for risk. Treating sepsis by machine learning algorithms is difficult due to a couple of reasons: There is limited and error-afflicted initial data in a highly complex biological system combined with the need to make robust, transparent and safe decisions. We demonstrate a suitable method that combines data imputation by a kNN model using a custom distance with state representation by discretization using clustering, and that enables superhuman decision-making using speedy $Q$-learning in the framework of distributional RL. Compared to clinicians, the recovery rate is increased by more than 3% on the test data set. Our results illustrate how risk-aware RL agents can play a decisive role in critical situations such as the treatment of sepsis patients, a situation acerbated due to the COVID-19 pandemic (Martineau 2020). In addition, we emphasize the tractability of the methodology and the learning behavior while addressing some criticisms of the previous work (Komorowski et al. 2018) on this topic.

## 1 Introduction

The present work addresses the treatment of sepsis using distributional reinforcement-learning (RL). The treatment of sepsis has enormous medical importance as it is a leading cause of death worldwide [3] which has also seen an increase in relevance as a complication in COVID-19 [4]. Sepsis treatment is also a grand challenge at the intersection of medicine and machine learning, where recent work underlined the importance of the use of data to improve existing medical knowledge [5, 6]. From a medical standpoint, it is highly safety-critical. From a machine learning perspective, despite the availability of a significant amount of unstructured data [7, 8], advanced and novel preprocessing techniques are required to make sense of the data. Moreover, sepsis treatment consists of modeling irregularly-sampled time series with long-term dependencies [9] which adds to its complexity.

**Funding:** This work was partly supported by FWF (Austrian Science Fund) START Project no. Y660 and by FWF (Austrian Science Fund) grant SFB F65. There was no additional external funding received for this study and received funds played no role in the research. The funders had no role in study design, data collection and analysis, decision to publish, or preparation of the manuscript.

**Competing interests:** The authors have declared that no competing interests exist.

Some recent works such as [10–12] focused on representing the policies as deep neural networks. However, the convergence properties of such algorithms in deep RL are largely unknown. Here, we employ algorithms in distributional RL whose convergence properties are much better understood. It is known that the *Q*-learning update rule converges to the fixed point of the composition of the Bellman operator and the categorical projection operator [13, 14], and that policy evaluation using the speedy *Q*-learning update converges to the return distribution [15] (see Section 3.1).

The overall quality of our decision-making algorithms is highly related to the quality and to the characteristics of the data. Biomarkers and physiomarkers suggested by recent works such as [16–18] can serve as additional sources of data that may enhance the performance of the algorithms, especially in cases where a more rapid action is required. We believe that combining the time-delayed data we used here with the data mentioned above collected by online testing can help develop even more performant sepsis diagnostic and treatment algorithms. In [17], a support-vector-machine (SVM) classifier was trained on data collected from active physiomarker sensor streams and successfully detected sepsis with a high rate. In the present work, the aim is not to perform sepsis diagnosis, but to learn and to evaluate treatment policies using distributional RL. Hence the present work is in principle complementary to [17] as the present algorithms can use the output of the SVM classifier as additional input for our decision-making agent to find better treatment policies.

[2] suggested a very promising increase of the treatment outcomes for sepsis. In particular, it was shown that time-critical treatment procedures can be reduced to the administration of vasopressors and intravenous fluids as acute measures, which proved to strongly correlated with the patient mortality. The work however, went under some methodological criticisms [19]: such as long time resolutions, modeling and assumptions of the transition dynamics, and the approach's interpretability. The authors have responded to some of the concerns in [20], however, the aforementioned concerns motivated us to improve over these aspects in the present study.

In particular, we decreased the time resolution to 60 minutes (4× faster than the method proposed in [2]), losslessly. This is a direct result of our improved scheme for the imputation of missing values described in Section 2.2. Furthermore, we claim that as the transition matrix intuitively depends on the states' approximated representation, brute-force search for the one that yields the best model performance [2] is ill-advised. In Section 2.3, we provide a method that takes the dynamics of the problem into account to select an effective state representation. Lastly, to address interpretability, we chose to implement a *distributional* RL algorithm, which models the distribution of the reward for each state-action pair instead of only the expected values. This provides more insights into the dynamics of the model in use, and yields a trustworthy decision-making tool compared to methods built on *classical* RL algorithms [13, 14]. Actions are represented by the volume of intravenous fluids and dose of vasopressors. The vasopressors consist of vasopressin, dopamine, epinephrine, norepinephrine and phenylephrine, while intravenous fluids consist of infusions of blood products, crystalloids, colloids and boluses. The state space is defined by using *k*-means clustering on time series of patient data represented with 53 features. (See supporting information for details.)

We emphasize that our approach is not aimed to modeling sepsis, but to provide a model for the clinicians' policies and calculate an optimal treatment strategy based on the successes and errors revealed in the clinical data. Besides data preparation, our algorithm does not require any prior knowledge about the biological processes involved, since it is (implicitly) taken into account through the statistical sampling of patient data. In addition to finding optimal policies for treatments, we design a novel *sepsis simulator* (see Section 2.4) that approximates the sepsis process while the patient is under treatment in the ICU.

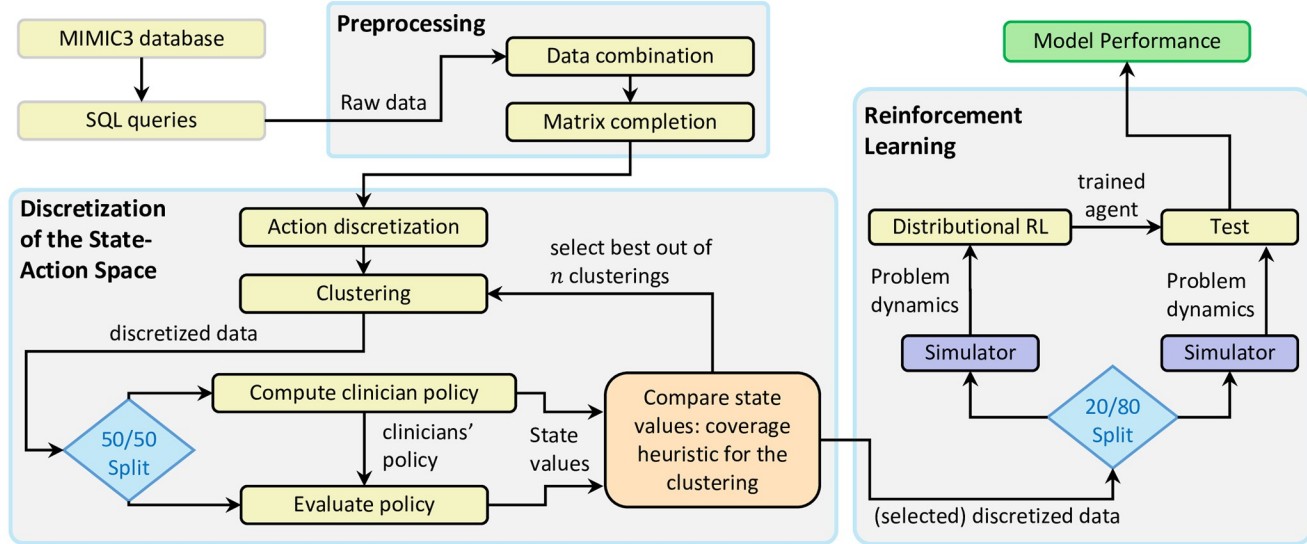

**Fig 1. Overall data flow.** The data flow including the three main challenges: preprocessing, state-action space representation, and reinforcement learning.

Our primary contributions can be summarized as: ensuring the traceability and quality of the data preparation process, developing an effective approach for the essential state-space approximation, and using distributional RL as the learning framework. This framework allows us to account for risk elements in a high-stakes medical decision-support system. Our sepsis treatment algorithm circumvents the concerns over the applicability of the previous methods and achieves super-human performance. To show the complexity of the method, we have illustrated our approach in Fig 1.

## 2 Historic data, methods, and algorithms

### 2.1 Dataset and data preparation

We based the data preparation process (taken from the MIMIC-III dataset, see [7]) on that described in [2], to ensure a fair comparison. This process consists of combining the features of interest followed by a matrix completion step to fill in missing values, verify the Sepsis-3 criteria [21] and run other algorithms. Access to the database was gained through PhysioNet, and emphasis was given to the responsible handling of patient data [22]. The dataset contains around 60 000 intensive-care unit (ICU) admissions and represents both demographic and clinical information on the patients in continuous or binary form. An approach mixing SQL queries and Julia programming [23] was chosen to properly organize data. To mitigate the long time horizon concerning other methods [19], we modified the time discretization from four-hour steps to one-hour steps to better model the biological processes under test. This, however, makes the missing-value imputation step more challenging. We address this by a novel method for matrix completion detailed in the following subsection.

### 2.2 Missing-value imputation

The dataset contains substantial amount of missing values which has to be taken care of. In the particular case of sepsis, we require some variables to be known to be able to apply the Sepsis-3 criteria [21] to decide whether a patient suffers from sepsis or not.

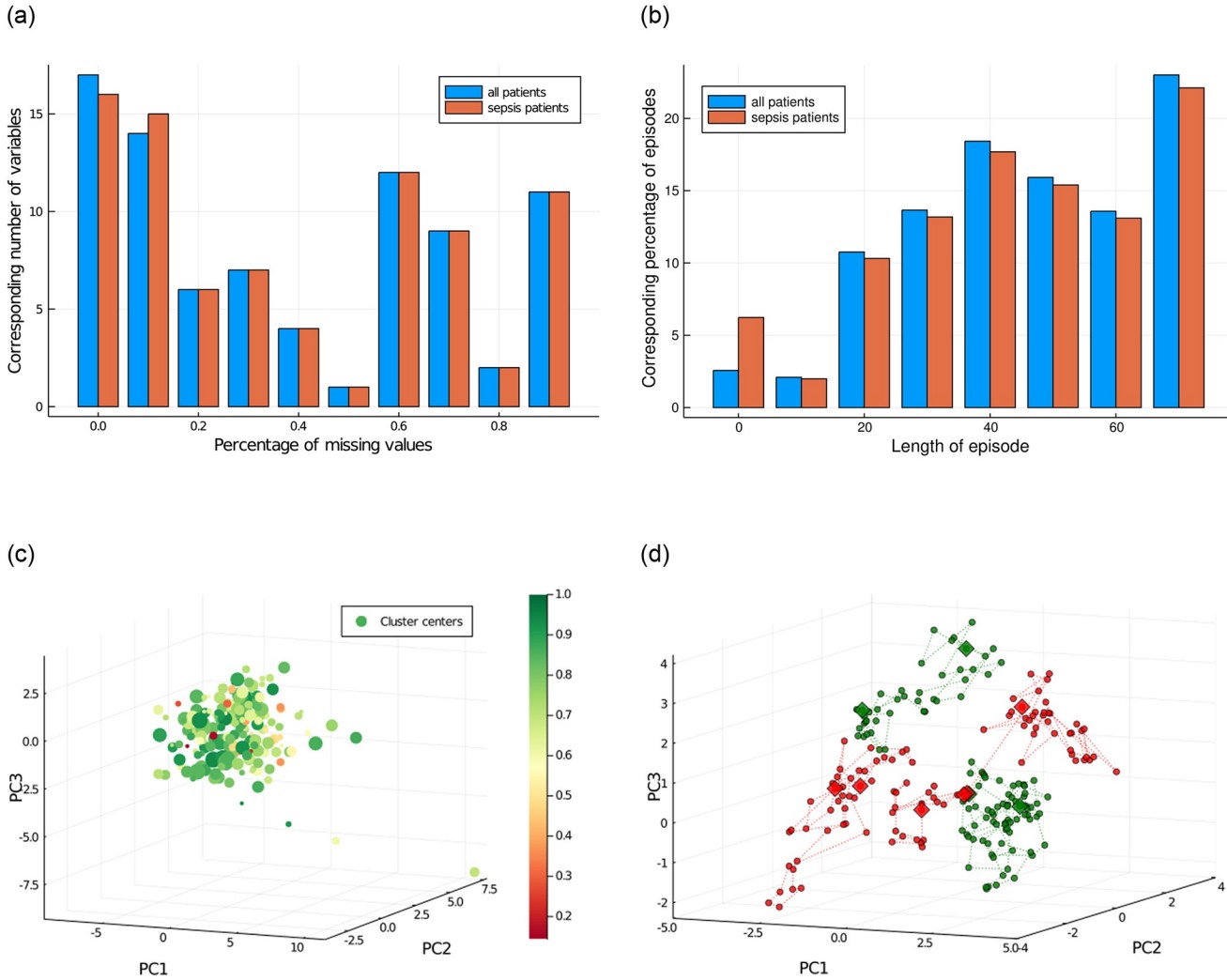

**Fig 2. Overview of the data. (a)** Histogram for the percentage of missing values before matrix completion. Features with more than 90% missing values were not kept. It appears that missingness does not depend on sepsis diagnosis. **(b)** Histogram for the length (i.e. number of timesteps) of the episodes. **(c)** 3D representation of cluster centers, using a linear projection along the first three principal components of the data. The size of the points is affine in the population of the clusters. The color scale corresponds to the survival rate in each cluster. **(d)** A representation of some episodes using the linear projection along principal components. We represent in green two episodes where the patient survived and in red three episodes where the patient died within 90 days.

In the imputation step of our data processing pipeline, we start with 957 563 observations each with 82 features. Among these features, 20 are not taken into account for this imputation step for they are either demographic features or binary. Among the remaining 62 features, we drop features that individually present more than 90% missing values and finally obtain 53 features subject to imputation, which present overall 39% of missing values (see Fig 2a).

All features contain missing-values. Because of this, a classical $k$-nearest-neighbors (kNN) model for imputation is not directly applicable since it requires a common ground between observations to compute pairwise distances (and gather nearest neighbors). [2] proposed to perform a linear interpolation step on features with a low percentage ($\leq 5\%$) of missing values—so that values are available for all observations—prior to the kNN algorithm that computes distances on the now common ground of complete features. However, this reveals to be sub-optimal, because there are only few features—3 in our implementation—that match the

low-percentage threshold criterion. Moreover, since values are sometimes missing because clinicians did not require them, all observations concerning one patient and their linear interpolation are ill-advised.

To overcome this problem, we introduce the following distance metric that is able to handle missing values and will replace the euclidean distance in the kNN computations. The distance is defined as:

$$D: \ \mathbb{R}^n \times \mathbb{R}^n \to [0, n), \ \ (x, y) \mapsto \sqrt{\sum_{i=1}^{n} f(x_i, y_i)},$$

where

$$f(x_i, y_i) := \begin{cases} \dfrac{(x_i - y_i)^2}{p_i + (x_i - y_i)^2} & \text{if } \ x_i, y_i \in \mathbb{R}, \\[3mm] a \in \left[\dfrac{1}{4}, 1\right] & \text{if } \ x_i \text{ or } y_i \text{ is missing,} \end{cases}$$

where $\{p_i\}_{1 \le i \le n}$ are positive scaling parameters and $a$ is a parameter that penalizes missing values: $a = 1$ corresponds to a "maximal" distance (in the sense of the function $f$) between known values and missing ones, whereas $a = \frac{1}{4}$ means that we prefer missing values to known far-away ones. The lower-bound $\frac{1}{4}$ is a necessary (and sufficient) condition for $D$ to satisfy the triangle inequality. This is an important feature, since this allows the use of efficient tree structures such as KD-trees and ball-trees to be used to be able to handle large amounts of data. We elaborate further on the performance of this method in a test-case of matrix completion, in the supplements. In this test-case, a Swiss-roll dataset (S1 Fig) is orthonormally mapped into a higher dimension such that resulting features depend on one another. We see in S4 Fig that our kNN method is effective, independent of the higher dimension in the sense that it outperforms the mean-imputation method that we use as a baseline in the matrix completion problem. Although singular-value thresholding (SVT) [24] outperforms kNN for the highest dimensions, it is very inefficient for lower dimensions. In that sense, the kNN method is more reliable. This motivates our choice of matrix completion method.

Moreover, we chose $a = 1$ and the $p_i$ to be twice the variance of the corresponding features. (This choice is motivated by the classical result $\mathbb{E}[(X - Y)^2] = 2V$ where $X$ and $Y$ are i.i.d. random variables and $V$ is their common variance.) Other advantages of this method are the low amount of parameters that are required and the available heuristics that allow straightforward choices. Other methods that require fine-tuning of sensitive parameters are not applicable in the present setting, because we do not have access to a complete, comparable dataset to train models on.

## 2.3 State representation

It is hard to find an approximation of the formations contained in the continuous raw data into a tabular form, specially due to high dimensionality. In this regard, the following problems arises:

- The amount of data is limited. Therefore, the train-test split of data faces a trade off, for avoiding overfitting. Hence, careful considerations has to be taken into account for efficient sampling.

- Edge cases and outliers in the data should not be ignored as they are special disease patterns.

- The K-Means clustering method does not provide a unique solution. Therefore, different cluster solutions emerge, which in our present work also show different performances in terms of learning behavior and representation of the state. The choice of state representation is critical for further procedure and the interpretation of the results. A substantial amount of useful information can be provided to the learning algorithm by an appropriate choice of cluster centers; it is of utmost importance.

Given these challenges, we aim to comply with the following additional conditions: first, we tend to avoid using importance sampling (which can be implemented in off-policy evaluation in general), as its deployment for patients records is not recommended: Since there is limited empirical data, one has to rely on off-policy methods, which can be learned by importance sampling and quantified by using off-policy evaluation. Unfortunately, this approach assumes that one knows the behavior policy well enough and that sufficient data is available. In the case of the present patient records, this is problematic, since importance sampling introduces a fundamentally higher variance and there is often too little data to reflect exact sequences in treatment [25]. As an alternative, approaches based on the generated model can be used, but their evaluation methods sometimes do not converge even with arbitrary amounts of data. [26]. Therefore we must choose a method that minimizes the biases that enter the procedure as much as possible. Second, the state representation should not anticipate learning performance, i.e., the state representation should preserve observed and possible behavior without performing any additional tasks. In other words, the representation should be independent of the learning algorithm used.

How can a suitable clustering be found? Our objective is to identify a clustering method that effectively approximates the continuous raw data, which have already been subdivided into one-hour windows. As a starting point, we have the 53 dimensional vectors that form our episode. We now want to make the best possible tabular case of this mass of data, and reduce each of these vectors to a number without destroying too much medical information. One method we use for this is KMeans similar to [2]: through this algorithm, clusters are searched in such a way that the value of features that are statistically correlated in the overall data end up in the same cluster.

However, there are at least two problems with this approach: First, it is not clear how many clusters should be determined (the algorithm requires this number as a parameter) in order to preserve the medical facts sufficiently well, but also to remain general enough to make meaningful statements. The second problem is that the algorithm is not deterministic and there is no unique solution, i.e., if the clustering is executed several times, one obtains a slightly different result each time. In the data preparation, we have run the algorithm 50 times each for 19 different numbers of clusters and obtain a total of 950 different ways of discretizing the raw data.

At this point, it is necessary to develop a meaningful decision criterion to select a viable solution. We refer to one method of implementation that we developed in this work as the "coverage heuristic", which can be described by the following considerations. A clustering solution is good if it preserves specific forms of actions and the resulting consequences well. Specifically, the actions of physicians are a good candidate here. In the reinforcement learning framework, numerical values are assigned to the respective states. In general, this is the expected value for the return. So we divide the available data into two halves and try to determine these values for the states under the clinician policy. For this we use a method from dynamic programming and obtain for 600 states, for example, 600 values of how good the respective states perform under this policy.

In the next step, we take the second half of the data and determine the values there as well, but now not with the "own" policy, but with the clinician behavior from the first half. This is the crucial step, because if it now turns out that bad states in the first half of the data are also bad states in the second half (or the same with good states), then this indicates that this clustering is obviously able to distinguish good and bad states.

We now use this idea to compare the quality of our 950 different solutions. To do this, we look at the respective halves of the data in pairs, determine their value under the clinican policy of the first half, and define the "coverage" as the relative frequency of numerical deviations smaller than 15. The value 15 is determined empirically and plausible, since the values range from −100 to 100.

We can summarize the procedure in the following steps:

1. We split the dataset into two halves and extract the physicians' policy from the first dataset.

2. We determine each state's value under the extracted policy using the classical approach via dynamic programming and policy evaluation.

3. We determine the values of the states of the second dataset under the above policy (calculating using the first dataset).

4. We determine the relative frequency of deviations in the state values from the first dataset smaller than 15 from the values from the second dataset and use the numeric value as a ranking for the quality of the approximation.

Using this quantitative approach, we can ensure that the discretization or clustering can represent a treatment policy sufficiently well. We thus obtain a tool to compare the quality of clustering for medical treatment among the variety of alternatives that can be calculated. It is important to note that we do not anticipate the policy that will be calculated later. This procedure depends exclusively on the unstructured data and the treatment histories of the patients. As a result, this decision heuristic could be called "end-to-end". It represents our selection criterion for processing the continuous data for use by the RL algorithm in the final step.

Any policy can be used to implement this procedure. The most promising option is to consider the existing clinicians' policies due to the limited amount of data. Initially, we used value iteration to compute each state's optimal values and chose these as the basis for comparison between any two halves of the dataset. This approach did not work well because of the limited amount of data available. Our comparison heuristic did not work because patient records in the two halves of the dataset were too different, and the supposedly optimal option in one dataset did not occur at all in the other. This motivated us to to choose the medics' policy as a reference.

## 2.4 The sepsis simulator

There are several possibilities for the interface between the state representation and the learning algorithm. Due to the characteristics of the environment off-policy methods are the most reasonable choice. For the evaluation of a learned policy, off-policy evaluation methods based on Importance Sampling, can be used [27]. Still, in the present application, it is hard to determine the value of deterministic strategies, and an analysis of the results is mainly limited to qualitative statements. As an alternative, one can use actor-critic methods [28] or extend off-policy evaluation methods [29].

We take a novel approach and develop a simulator that generates new trajectories from the existing transition probabilities for the interface between the preprocessed trajectories and the

learning algorithm. This platform ensures a clean separation between the environment and agent, while evaluates the generated data more comprehensively.

However, there were two challenges with the implementation. Since not all 25 actions (two drugs and five doses) are available in every state, we had to reduce the action selection to valid actions. This adds complexity, but does not contradict the framework of Markov decision processes (MDPs). The second problem was loops, where the agent would be caught, and the episodes did not terminate. The cause leads to a strong correlation with the quality of the state representation, and the experience gained from preventing loops also served as a refinement of our methodology. The number of clusters also played a crucial role, illustrating the impact of overfitting.

The simulator is initialized with 20% of the dataset prepared in the form of trajectories for training. It can interact with the agent via an OpenAI gym [30] compatible interface. The individual patient records represent the episodes; a reward of + 100 (survival) or −100 (death) is only given at the ends of the episodes. In each step, the agent is informed about the possible actions and decides according to the learned policy (see Section 3.1).

# 3 Distributional reinforcement learning

The typical approach to reinforcement learning is to model the expected return by either a state value function $v$ or a state-action value function $q$. As the name suggests, the core of distributional reinforcement learning is to model the entire distribution of the return instead of only its expected value.

For $(x, a) \in \mathcal{X} \times \mathcal{A}$ the *return* $Z^\pi(x, a)$ is the sum of the discounted rewards along a trajectory following a policy $\pi$ starting in state $x$ and taking action $a$, i.e.,

$$Z^\pi(x, a) := \sum_{t=0}^{\infty} \gamma^t R(X_t, A_t),$$

$$X_0 := x, \ A_0 := a, \ X_{t+1} \sim p(\cdot|X_t, A_t), \ A_{t+1} \sim \pi(\cdot|X_{t+1}).$$

The function $Z^\pi$ mapping state-action pairs to random variables is called the *return distribution function*.

We can relate the state-action value function to the return distribution function by observing that $q(x, a) = \mathbb{E}[Z^\pi(x, a)]$. Furthermore, the Bellman equation can be extended to the distributional case as

$$Z^\pi(x, a) \stackrel{D}{=} R(x, a) + \gamma Z^\pi(X', A'),$$

where $X' \sim p(\cdot|x, a)$, $A' \sim \pi(\cdot|X')$. Here the equal sign indicates that the random variable on the left-hand side and the one on the right-hand side are identically distributed.

Lastly, $\eta_\pi^{(x,a)}$ denotes the underlying probability distribution of the random variable $Z^\pi(x, a)$, giving us a second representation

$$Z^\pi(x, a) \sim \eta_\pi^{(x,a)}$$

of return distribution functions.

## 3.1 Categorical distributional reinforcement learning

The task of *policy control* is to find the return distributions $\eta^*$ which are optimal with respect to the expected value $q(x, a) = \mathbb{E}_{Z \sim \eta_*^{(x,a)}}[Z]$. As we are interested in the entire return

distribution, the problem of finding an approximation method might arise. We decided to consider distributions over fixed finite support $z_1, \ldots, z_N$. This method is well-known and has the advantage of being highly expressive and computationally tractable [13].

The set of *categorical distributions* is defined as

$$\mathcal{P}_z := \left\{ \sum_{i=1}^{N} p_i \delta_{z_i} \;\middle|\; p_i \geq 0, \;\; \sum_{i=1}^{N} p_i = 1 \right\}.$$

With this notation the return distribution of any policy $\pi$ can be modeled as $\eta^{(x,a)} \in \mathcal{P}_z \approx \eta_\pi^{(x,a)}$ for all state-action pairs $(x, a) \in \mathcal{X} \times \mathcal{A}$.

Under the Bellman operator $\mathcal{T}^\pi$, the support of a distribution $v \in \mathcal{P}_z$ changes and in general $\mathcal{T}^\pi \eta^{(x,a)} \notin \mathcal{P}_z$. Therefore, one has to perform a projection back onto the support $z_1, \ldots, z_N$. This can be done using the *categorical projection operator* $\Pi_C$ such that $\Pi_C \mathcal{T}^\pi \eta^{(x,a)} \in \mathcal{P}_z$. For more details of this method, see [13, 14].

Instead of the *Q*-learning [31] update rule

$$\eta_{k+1}^{(x,a)} = (1 - \alpha_k)\eta_k^{(x,a)} + \alpha_k \Pi_C \mathcal{T}^{\pi_k} \eta_k^{(x,a)},$$

we opted for the SQL update rule [32]

$$\eta_{k+1}^{(x,a)} = \eta_k^{(x,a)} + \alpha_k \left( \Pi_C \mathcal{T}^{\pi_k} \eta_{k-1}^{(x,a)} - \eta_k^{(x,a)} \right) + (1 - \alpha_k)(\Pi_C \mathcal{T}^{\pi_k} \eta_k^{(x,a)} - \Pi_C \mathcal{T}^{\pi_k} \eta_{k-1}^{(x,a)}), \qquad (1)$$

where $\pi_k$ is the policy which is greedy with respect of the expected values of $\eta_k$.

Reference [14] proved convergence of the *Q*-learning update rule to the fixed point of $\Pi_C \mathcal{T}^{\pi_*} : \mathcal{P}_z^{\mathcal{X} \times \mathcal{A}} \to \mathcal{P}_z^{\mathcal{X} \times \mathcal{A}}$ for an optimal policy $\pi^*$ (w.r.t. the expected value) denoted by $\eta^*$ in the maximum Cramér distance

$$\begin{aligned}
\bar{\ell}_2(\eta, \xi) &:= \sup_{(x,a) \in \mathcal{X} \times \mathcal{A}} \left( \int_{\mathbb{R}} |F_{\eta^{(x,a)}}(z) - F_{\xi^{(x,a)}}(z)|^2 dz \right)^{1/2} \\
&= \sup_{(x,a) \in \mathcal{X} \times \mathcal{A}} \left( \sum_{i=1}^{N-1} (z_{i+1} - z_i)(F_{\eta^{(x,a)}}(z_i) - F_{\xi^{(x,a)}}(z_i))^2 \right)^{1/2}.
\end{aligned}$$

It can be shown that using the SQL update rule, the accelerated convergence is proven in the expected-value case also holds in the distributional case when policy evaluation is used to find the return distribution of a given policy $\pi$ by keeping $\pi_k = \pi$ fixed [15]. Empirical results show that the same accelerated performance is also achieved for the SQL update rule (1) when used for policy control.

Further, suppose we decrease the maximum distance of fixed atoms (i.e., by increasing the number of equally spaced atoms). In that case, the categorical approximation $\eta^*$ is closer to the true return distribution $\eta_{\pi_*}$ in terms of the Cramér distance.

Finally, the simulator allows us to perform updates in a *synchronous* fashion. That is, in each iteration, $\eta_k$ is updated at *every* state-action pair. Thereby the problem of exploring the state space, which we would have if the agent is trained in an online fashion, is avoided, and convergence to the approximated return distribution function $\eta_\pi$ is faster.

The described method is summarized in Algorithm 1. As already mentioned, the algorithm can be easily converted to policy evaluation by keeping the policy fixed $\pi_k = \pi$ and sampling $a_k' \sim \pi(\cdot | x_k')$ in line 8 instead of using the greedy action with respect to the expectation.

**Algorithm 1** Synchronous Speedy Categorical Policy Control

```
1: Require: η_k^(x,a) = Σ_{i=1}^N p_{k,i}^(x,a) δ_{z_i} for fixed atoms z_1, ...z_N
2: Input: discount factor γ, max. number of iterations T, initial
   guess η_0, threshold Δ
3: η_{-1} ← η_0
4: for k ∈ 0, ..., T - 1 do
5:     α_k ← 1/(k+1)
6:     π_k(x) ← arg max_{a∈A} E_{Z~η_k^(x,a)}[Z]  ∀x ∈ X
7:     for (x,a) ∈ X × A do
8:         Sample x'_k ~ p(·|x,a), a'_k = π_k(x'_k), r_k ~ R(x, a)
9:         T_k^{π_k} η_k^(x,a) ← Σ_{i=1}^N p_{k,i}^(x'_k,a'_k) δ_{r_k+γz_i}  # Bellman update
10:        T_k^{π_k} η_{k-1}^(x,a) ← Σ_{i=1}^N p_{k-1,i}^(x'_k,a'_k) δ_{r_k+γz_i}  # Bellman update
11:        # Project onto support z_1, ..., z_N and calculate difference
12:        D_k^(x,a) ← kΠ_C T_k^{π_k} η_k^(x,a) - (k-1)Π_C T_k^{π_k} η_{k-1}^(x,a)
13:        # Update η
14:        η_{k+1}^(x,a) ← (1 - α_k)η_k^(x,a) + α_k D_k^(x,a)
15:    end for
16:    if ℓ̄_2(η_{k+1}, η_k) ≤ Δ then
17:        break
18:    end if
19: end for
```

# 4 Results

For all applications of the speedy categorical algorithm, 51 equally spaced atoms on the interval $[-100, 100]$ were used. As the overall outcome is more desirable than the immediate rewards, the discount factor was set to $\gamma = 0.99$. We allowed a maximum number of thousand iterations, i.e., $T \coloneqq 1000$. For policy control, the threshold was set to $\Delta \coloneqq 0.05$, whereas, for policy evaluation, we did not break the loop early.

## 4.1 Generalization problems

In order to assess the performance of the resulting policies, we split the dataset into training (80%) and test (20%) sets and built simulators for both sets separately. We observed that the state transition probabilities of both simulators could be quite different. As a result, an action that is good in one simulator can be bad in the other one. It is expected that if more data is available, the transition probabilities will be more similar.

In our experiments, the differences in transition probabilities resulted in a deterministic policy that performs (near) optimal in the training simulator to generate trajectories that loop back to already visited states. Using a stochastic policy that chooses the best action with probability 0.5, the second-best with probability 0.25, and so on avoids the generation of non-terminating trajectories and is therefore used for performance assessment of the agents.

For several states, we repeated policy control 10 times for random train-test splits. The results are shown in Fig 3. The recovery rate was estimated by simulating 10 000 patient trajectories starting from randomly selected initial states. Even using these stochastic policies, the gap between the performance using the training and the test simulator is substantial across different number of states.

On the training set, the computed policy outperforms the clinicians' policy with recovery rates near one for numbers of states greater than 600. In contrast, on the test dataset, the recovery rates drop to the clinicians' level of about 90%. It can also be seen that for numbers of states below 600, the state space becomes too coarse to find a policy with (near) perfect performance.

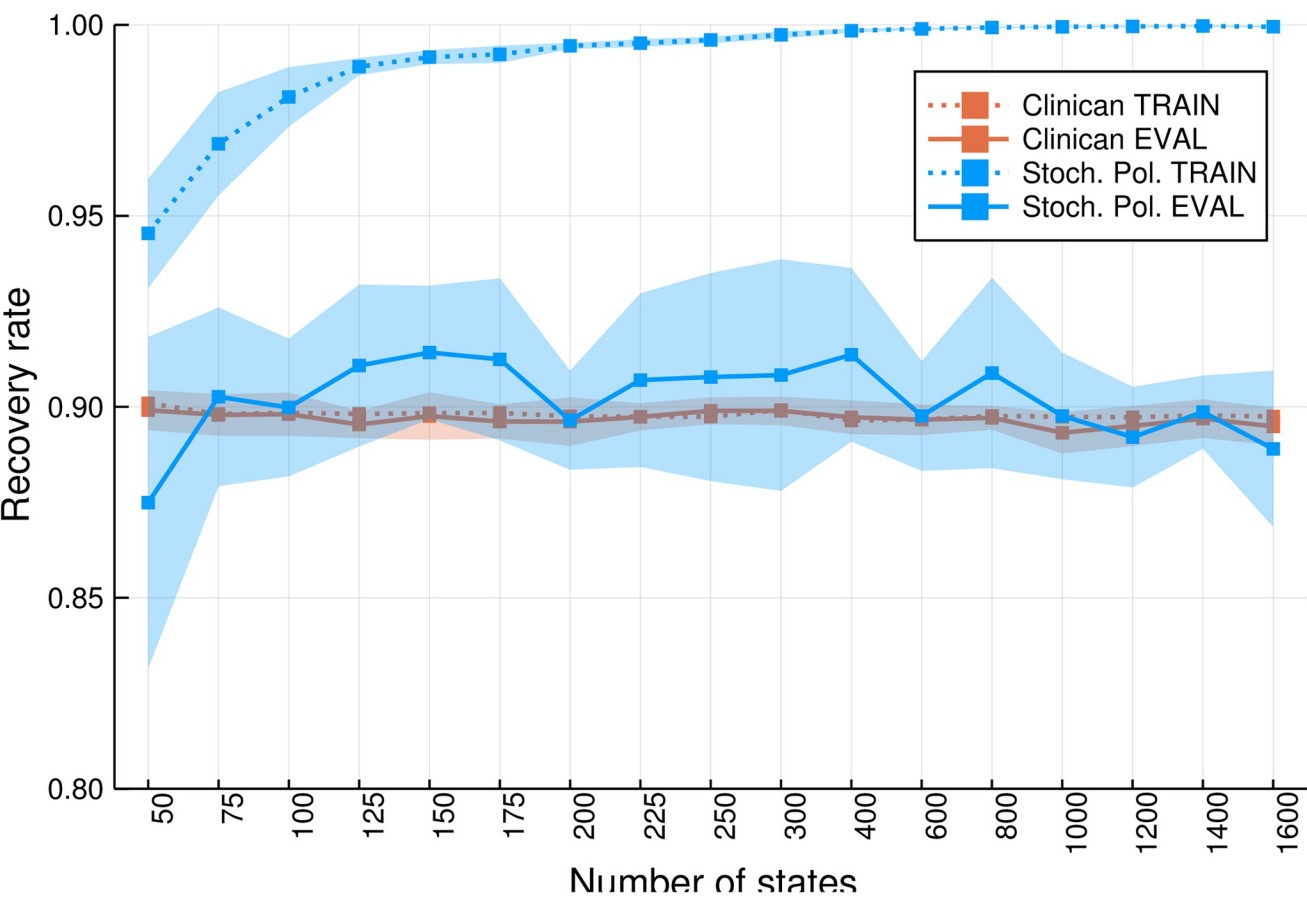

**Fig 3. Comparison of the performance of stochastic policies on the training and test simulators for different numbers of states.**

Again, we emphasize that we expect the gap to shrink if more data become available. Another way to overcome this generalization problem would be to include the learning of the state approximation or representation into the learning algorithms, which could be the subject of future research.

### 4.2 Clinicians' policy using 600 states

The clinicians' policy is the stochastic policy that takes actions with probabilities proportional to the number of times clinicians took action for any given state in the data.

When evaluating all available patient trajectories, results show that less than 0.05% of the clinicians' actions predominantly lead to a negative outcome (defined as a probability larger than 50% of the patient dying). About 0.3% of the actions had a highly bimodal return distribution. These are actions where a positive outcome is almost as likely as a negative outcome. The majority of actions ($\approx$ 96%) predominantly led to recovery and had a non-negligible probability (5% to 20%) of a negative outcome. With a recovery probability over 95%, around 0.2% of all actions could be considered completely safe. In Fig 4, examples of the various return distributions discussed here can be seen.

### 4.3 Optimal agent using 600 states

Here, we investigate the optimal agent, which was obtained based on 600 states by performing Algorithm 1 on a randomly chosen train-test split with the parameters setup discussed above.

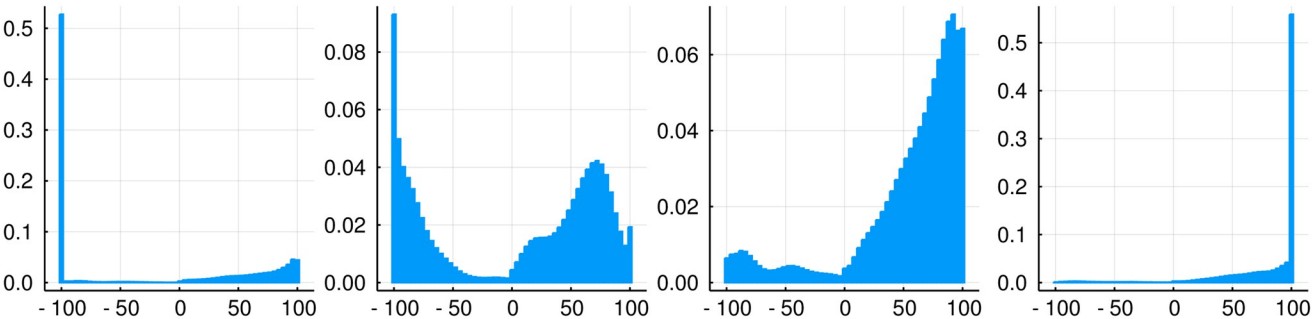

**Fig 4. Examples of return distributions when following the clinicians' policy.** From left to right: negative outcome, bimodal distribution, slightly bimodal distribution, positive outcome.

As shown in Fig 3, 600 is the minimum number of states for which a recovery rate of approximately one was achieved on the training set on average.

In Fig 5, we observe the approximated return distributions for each initial state choosing actions according to the stochastic policy discussed previously. Surprisingly, it is possible to select actions such that from every initial state, the recovery of the patient is guaranteed with very little variance in treatment length. For almost every initial state, the return distribution is left-skewed, corresponding to fast patient recovery. However, this perfect performance was only achieved on the data the agent was trained on.

In order to test the performance of the calculated policy, 10 000 episodes were simulated in the test simulator. The results are shown in Tables 1 and 2. Using the test simulator, the optimal agent still outperformed the clinicians' policy comparing the 90-day mortality.

## 5 Discussion, scope and conclusions

The results advocate for the effectiveness of our distributional RL methods for finding good policies. Since we used the expected return as the maximization objective, the approach is quite similar to optimizing the expected return directly like in standard RL methods. It is however hypothesized in [13] that using distributional RL allows for a better representation in the context of function approximation (present here through the states' discretization). Having the return distribution at hand also allows for many different maximization objectives (for example, penalizing variance or penalizing risk), but they were not studied in the present work.

An important assumption of our method is that the process at stake can be modeled by a piece-wise constant function (see Section 2.3). Mathematically, this relates to the function to be approximated being *absolutely continuous*. In practice, this implies that the clustering needs a finer resolution where the function to approximate has an important gradient to produce a more accurate model. However, this modeling restriction has great benefits in regards to interpretability since any dependency on the features is immediate and explicit in our state representation.

Additionally, the spatial coherence induced by the chosen method for clustering allows for a neat representation of the high-dimensional data. We show some examples in Fig 6. Using PCA, the continuous features are linearly projected onto a two-dimensional plane in Fig 6a. We then used stereographic projection in Fig 6b through Fig 6d to produce a bounded representation of the projected state space.

In these figures, a certain state, parameterized by the center of the sphere, is mapped to the south pole of the sphere. The bottom half of the sphere then represents a neighborhood of this

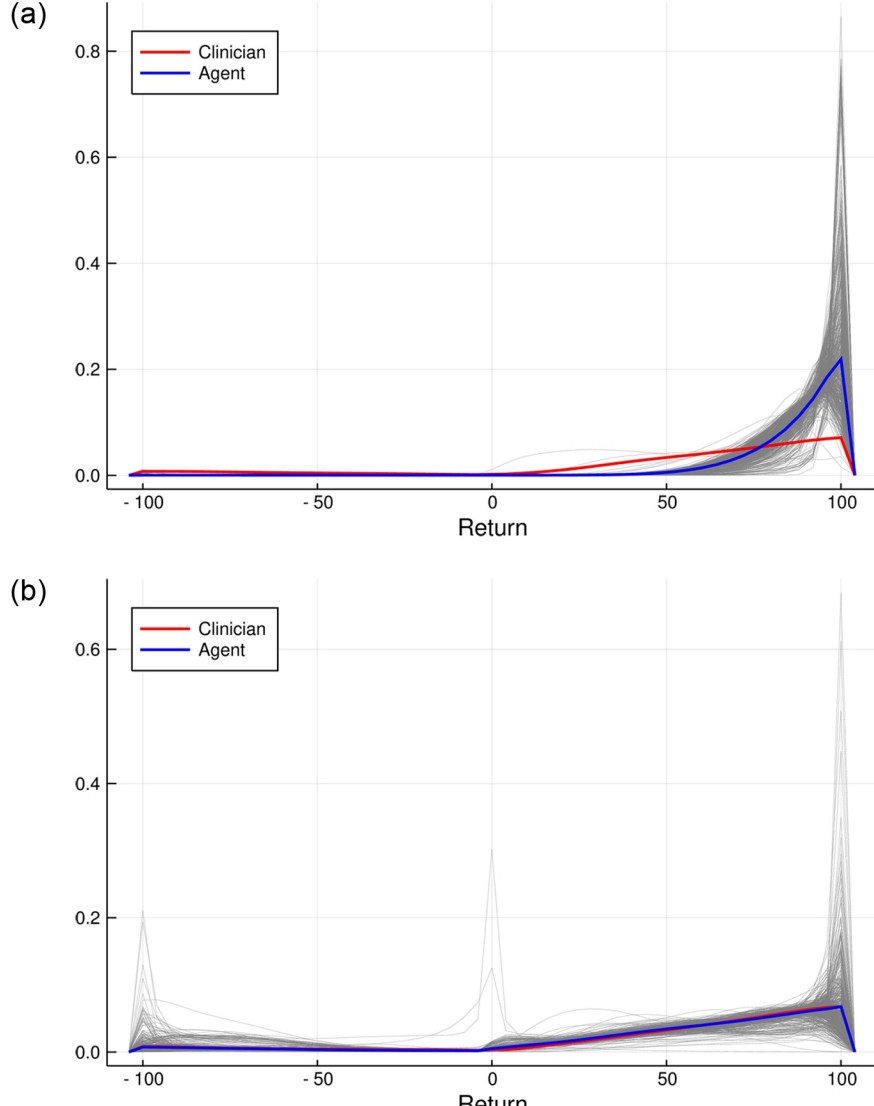

**Fig 5. Return distributions of the resulting stochastic policy for all initial states in gray and average return distribution (weighted by the numbers of occurrences as initial states) in blue.** The average return distribution of the clinicians' policy is shown in red for comparison. Left: training simulator; right: test simulator.

state parameterized by the specified radius. States outside of this neighborhood are mapped to the upper half of the sphere, with states infinitely far away from the center being mapped to its north pole. This new state representation can then be colored following results from the algorithm. We represent state-values (i.e. maximal expected $q$-values across actions) in Fig 6b and

**Table 1. Performance comparison of the agent and the clinicians after 10 000 simulated patient trajectories in the evaluation environment ($n$ = 10 000, 90-day mortality).**

|  | ENV | Mean reward | Recovery rate |
|---|---|---|---|
| Clinician | test | 47.47 | 85.41 |
|  | train | 51.77 | 88.47 |
| Agent (stochastic policy) | test | 50.83 | **88.76** |
|  | train | 86.84 | 99.80 |

**Table 2. Performance comparison of the agent and the clinicians after 10 000 simulated patient trajectories in the evaluation environment ($n$ = 10000, 28-day mortality).**

| | ENV | Mean reward | Recovery rate |
|---|---|---|---|
| Clinician | test | 54.13 | 89.57 |
| | train | 53.98 | 89.63 |
| Agent (stochastic policy) | test | 53.94 | 87.08 |
| | train | 91.29 | 99.95 |

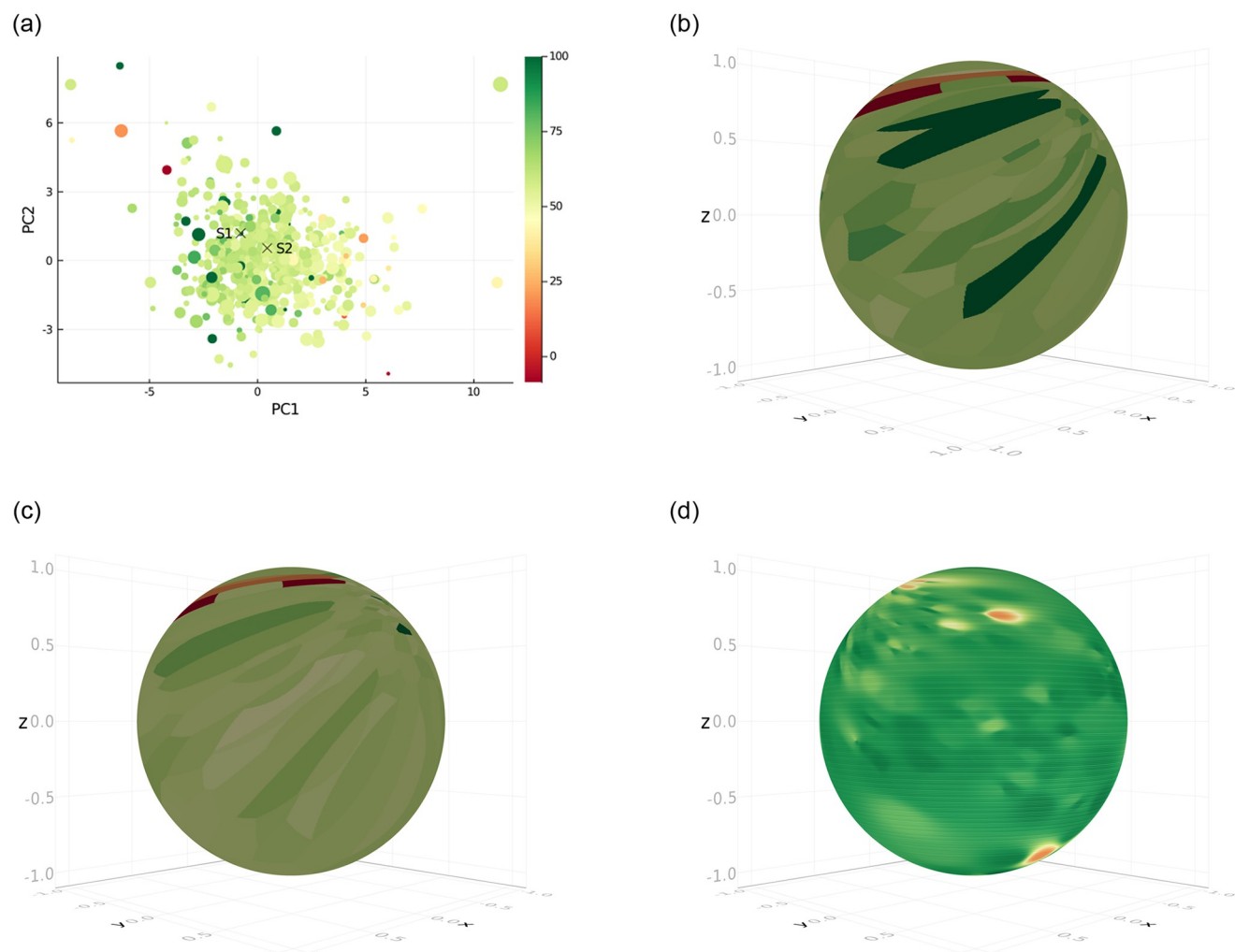

**Fig 6. Stereographic representation of the data produced by the model. (a)** Two-dimensional representation of the computed state values for the clinicians' policy using PCA. Each dot is a state obtained through the method described in Section 2.3 with the size matching the number of corresponding observations in the dataset. The coloring corresponds to the computed state value. $S_1$ and $S_2$ are specific states from which stereographic representation is produced. **(b)** (Inverse) stereographic projection from the PCA plane to a sphere with center $S_1$ and radius 1. The coloring corresponds to the state values. **(c)** The coloring is changed to fit the computed state-action values for some specified action. **(d)** The sphere is now centered at $S_2$ and the coloring corresponds to a different action. The state-action values were smoothed using inverse distance weighing [33] instead using of piece-wise constant approximation.

state-action values in Fig 6c and 6d for different configurations. Note the red spot near the south pole of the sphere in Fig 6d which indicates that taking the considered action resulted in a poorer outcome in a neighboring state than the state at the south pole.

This representation allows for a complete overview of the state space and can help overcome the limitation of the tabular approximation. Finally, we underline that such a representation which give rise to better interpretability would not be feasible for deep RL methods [10–12].

In Fig 6d, we also smoothed the state-action values using fuzzy membership functions instead of clusters. This stems from the idea that the K-Means algorithm used for clustering and state representation can be generalized using the fuzzy C-Means algorithm [34].

We also note that, although the biological understanding of sepsis mechanisms has not been the goal of this modeling approach, the data-driven tools (see Fig 2c and 2d) and models of treatment strategies can be exploited in future work to better understand sepsis and test its mathematical modeling.

We would like to add a few remarks about the limitations of our method. Of course, the performance of the learning method depends on the quality of the underlying data. We would also like to mention that our approach is not intended to diagnose sepsis, but to treat it. It turns out that reinforcement learning is an excellent choice in this context, because conclusions can be drawn for prior actions even from much delayed reactions of the environment.

From our point of view, it is also very promising from a methodological point of view, since this approach is more holistic than the pure classification and application of predefined strategies. In essence, our approach allows also a more fine-grained sequence of actions that can be extracted from existing patient records and treatment histories. The use of distributive RL makes sense from our point of view, since in further research steps one should of course also be able to quantify the risk of actions.

Regarding a further limitation, as with all machine learning problems, there remains of course the question of how much domain knowledge one may and should put into it. One point where care is needed here is clearly the reward function. In order to make the maximum possible valid statement for this problem, we intentionally decided against assigning intermediate rewards, i.e., whether patients survived or not is the only criterion. Regarding the issue how long the time interval of survival after ICU treatment should be to decide that question, we found different information in the literature. Due to the availability of raw data, we used the survival status after 90 days, which should be viable for the purposes of treating sepsis.

## Supporting information

**S1 Fig. The original 3D Swiss-roll dataset.** It is mapped into a 100-dimensional space using a random orthonormal mapping. (We applied the Gram-Schmidt process on a uniformly random matrix.) We then uniformly removed 70% of the observations and performed matrix completion using imputation by the mean observed value of features, kNN, and SVT [24]. (TIF)

**S2 Fig. Data recovered when imputing missing values using the mean value of the corresponding feature.** This method serves as a baseline for other completion methods. (TIF)

**S3 Fig. Data recovered when imputing missing values using a kNN algorithm based on our custom metric.** The general structure of the original data is well recovered. (TIF)

**S4 Fig. Comparison between mean imputation, kNN, and SVT for multiple dimensions of the orthonormally mapped data.** Although SVT gives an almost perfect recovery for very high dimensions, it performs very poorly (worse than mean imputation!) in low dimensions. On the other hand, the kNN recovery is always better (in terms of RSSE) than the mean imputation.
(TIF)

**S1 Table. Medical features extracted from raw data.**
(XLSX)

## Author Contributions

**Conceptualization:** Daniel Pasterk, Ramin Hasani, Clemens Heitzinger.

**Data curation:** Julien Malle, Hrvoje Kukina.

**Formal analysis:** Julien Malle, Hrvoje Kukina.

**Funding acquisition:** Clemens Heitzinger.

**Investigation:** Markus Böck, Julien Malle.

**Methodology:** Markus Böck, Julien Malle, Daniel Pasterk, Ramin Hasani.

**Project administration:** Daniel Pasterk, Ramin Hasani.

**Resources:** Daniel Pasterk.

**Software:** Markus Böck, Daniel Pasterk, Hrvoje Kukina.

**Supervision:** Ramin Hasani, Clemens Heitzinger.

**Validation:** Markus Böck, Daniel Pasterk.

**Visualization:** Markus Böck, Julien Malle, Daniel Pasterk.

**Writing – original draft:** Markus Böck, Julien Malle, Daniel Pasterk, Ramin Hasani, Clemens Heitzinger.

**Writing – review & editing:** Daniel Pasterk, Ramin Hasani, Clemens Heitzinger.

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
