## [Decision Letter · Decision Letter 0]

6 Jun 2021

PONE-D-21-12126

Superhuman Performance for Sepsis Treatment by Distributional Reinforcement Learning

PLOS ONE

Dear Dr. Pasterk,

Thank you for submitting your manuscript to PLOS ONE. After careful consideration, we feel that it has merit but does not fully meet PLOS ONE’s publication criteria as it currently stands. Therefore, we invite you to submit a revised version of the manuscript that addresses the points raised during the review process.

We look forward to receiving your revised manuscript.

Kind regards,

Chi-Hua Chen, Ph.D.

Academic Editor

PLOS ONE

Journal Requirements:

PLOS requires an ORCID iD for the corresponding author in Editorial Manager on papers submitted after December 6th, 2016. Please ensure that you have an ORCID iD and that it is validated in Editorial Manager. To do this, go to ‘Update my Information’ (in the upper left-hand corner of the main menu), and click on the Fetch/Validate link next to the ORCID field. This will take you to the ORCID site and allow you to create a new iD or authenticate a pre-existing iD in Editorial Manager. Please see the following video for instructions on linking an ORCID iD to your Editorial Manager account: https://www.youtube.com/watch?v=_xcclfuvtxQ

This work was partly supported by FWF (Austrian Science Fund) START Project no.

Y660 and by FWF (Austrian Science Fund) grant SFB F65. R.H. is supported by Boeing.

4. Please ensure that you refer to Figure 1 in your text as, if accepted, production will need this reference to link the reader to the figure.

Reviewers' comments:

Reviewer's Responses to Questions

**Comments to the Author**

1. Is the manuscript technically sound, and do the data support the conclusions?

Reviewer #1: Yes

Reviewer #2: Partly

2. Has the statistical analysis been performed appropriately and rigorously? 

Reviewer #1: Yes

Reviewer #2: I Don't Know

3. Have the authors made all data underlying the findings in their manuscript fully available?

Reviewer #1: Yes

Reviewer #2: No

4. Is the manuscript presented in an intelligible fashion and written in standard English?

Reviewer #1: Yes

Reviewer #2: Yes

5. Review Comments to the Author

Reviewer #1: Superhuman Performance for Sepsis Treatment by Distributional Reinforcement Learning

The authors propose a speedy implementation of a reinforcement algorithm for sepsis. In response to Komorowski et al;. 2018, initial work, the authors propose some changes which they believe increases the performance of sepsis algorithms. The descriptions of state representation and the sepsis simulator are clear and concise. Similarly, the explanation of distributional Q-learning is very thorough and comprehensible. However there are concerns that should be addressed.

Comments:

1. Title seems inappropriate unless the authors are reporting a randomized clinical trial demonstrating such ‘superhuman’ performance, I would suggest sticking with a more modest terminology.

2. The decreased time-resolution to 60 minutes is nebulous and unjustified. Considering that labs, vitals, medication and other critical variables arrive at frequencies in the range of 120-250 minutes, the narrow time resolution makes me believe this algorithm is tuned to identify acute deterioration rather than sepsis. If the authors believe otherwise, please include clinical justifications.

3. Figure 2 is difficult to follow, please characterize missingness as a function of sepsis diagnosis, it’d be interesting to see this distribution among the features that were kept out.

4. Figure 2d should be recharacterized, patients that have an in-hospital death within 90 days could die of any reason, not necessarily related to sepsis. Consider in-hospital death by day 21, as it is more likely to be associated with the disease.

5. Did you evaluate whether the missingess was at random or not-at-random, i.e. did your sepsis patients have a greater prevalence of these measures than your controls, if this was the case how did you address this characterization?

6. Please list the 53 variables that we retained after filtering for missingness.

7. Figure 7a-d is missing, should probably say supplemental 1.

8. It is a little unclear how the quantitative decision heuristic is used in determining the quality of a specific clustering profile. A figure illustrating the heuristic process in addition to the explanation would help to further elucidate the cluster quality heuristic.

9. Have the authors considered incorporating any short-term rewards into the simulated environment in order to improve the agent’s results. For instance, perhaps there could be some reward that increases an agent’s score for earlier and accurate diagnoses of sepsis.

10. Since a bulk of the data used in this work is derived from time-delayed measures, the authors should discuss recent similar work in the sepsis space that utilize sensor-derived physiomarkers to see how those approaches can be used to improve the state definitions. Please see:

Alqahtani, M.F., Marsillio, L.E. and Rozenfeld, R.A., 2014. A review of biomarkers and physiomarkers in pediatric sepsis. Clinical Pediatric Emergency Medicine, 15(2), pp.177-184.

Mohammed, Akram*; Van Wyk, Franco†; Chinthala, Lokesh K.*; Khojandi, Anahita†; Davis, Robert L.*; Coopersmith, Craig M.‡; Kamaleswaran, Rishikesan‡ Temporal Differential Expression of Physiomarkers Predicts Sepsis in Critically Ill Adults, SHOCK: September 28, 2020 - Volume Publish Ahead of Print - Issue - doi: 10.1097/SHK.0000000000001670

Zimmet, A.M., Sullivan, B.A., Moorman, J.R., Lake, D.E. and Ratcliffe, S.J., 2020. Trajectories of the heart rate characteristics index, a physiomarker of sepsis in premature infants, predict Neonatal ICU mortality. JRSM cardiovascular disease, 9, p.2048004020945142.

11. The limitations section of the manuscript is lacking, there are many limitations in this approach and it should be reported clearly.

Reviewer #2: To improve the Sepsis Treatment for clinicians, a machine learning method, distributional reinforcement learning, is proposed in this study. The agent with the proposed method performed higher efficiency comparing to the human case. The idea and the method is very interesting, and experiment results are valid. However, I suggest authors to improve the manuscript considering the follows.

1. The definition of problem needs to be given more clearly. In fact, readers like a clinician or an AI researcher are confused by the novel idea. My first question is what the input space of reinforcement learning is, i.e., the concrete states, and what actions are?

2. How to evaluate the proposed method? Is it better than other reinforcement learning models? To answer this question, experiment results are needed.

Thanks for your great effort in this study.

6. PLOS authors have the option to publish the peer review history of their article (what does this mean?). If published, this will include your full peer review and any attached files.

Reviewer #1: No

Reviewer #2: No

---

## [Author Response · Author response to Decision Letter 0]

28 Jul 2021

Detailed answers can be found in the external file "PlosOne_Answer_Reviewers.pdf"

---

## [Decision Letter · Decision Letter 1]

16 Aug 2021

PONE-D-21-12126R1

Superhuman Performance for Sepsis Treatment by Distributional Reinforcement Learning

PLOS ONE

Dear Dr. Pasterk,

Thank you for submitting your manuscript to PLOS ONE. After careful consideration, we feel that it has merit but does not fully meet PLOS ONE’s publication criteria as it currently stands. Therefore, we invite you to submit a revised version of the manuscript that addresses the points raised during the review process.

We look forward to receiving your revised manuscript.

Kind regards,

Chi-Hua Chen, Ph.D.

Academic Editor

PLOS ONE

Reviewers' comments:

Reviewer's Responses to Questions

**Comments to the Author**

1. If the authors have adequately addressed your comments raised in a previous round of review and you feel that this manuscript is now acceptable for publication, you may indicate that here to bypass the “Comments to the Author” section, enter your conflict of interest statement in the “Confidential to Editor” section, and submit your "Accept" recommendation.

Reviewer #1: (No Response)

2. Is the manuscript technically sound, and do the data support the conclusions?

Reviewer #1: (No Response)

3. Has the statistical analysis been performed appropriately and rigorously? 

Reviewer #1: Yes

4. Have the authors made all data underlying the findings in their manuscript fully available?

Reviewer #1: Yes

5. Is the manuscript presented in an intelligible fashion and written in standard English?

Reviewer #1: Yes

6. Review Comments to the Author

Reviewer #1: Thank you for your response. However I think you should specifically address the relevant citations in your discussions, the lack of such related works is not appropriate at this stage of the sepsis AI research.

1. Can you clarify how missingness was computed, i.e. at the population level or patient level?

2. Figure 2b can be separated into sepsis and controls to see whether there are any differences.

3. Figure 2c and 2d are example patients, why were these patients selected and how is this representative of the entire sample size?

4. I'm a bit confused on mortality/survival, why was 90 days chosen? Patients who died 90 days post meeting sepsis criteria may not have died of sepsis directly, how do you ascertain this? This needs some further investigation and justification.

5. Again the title is inappropriate, there are no treatments (i.e drugs) for sepsis, there are only therapies (fluid resuscitation, vasoconstriction, etc.) that can reduce mortality.

6. The outcome measure is also somewhat nebulous, if you define successful management as reduction in death alone, that is problematic as you in-hospital mortality statistics are wholly inappropriate for determining outcomes. I would consider you expand your outcome definition to look at concrete clinical measures, i.e. successful extubation, successful resuscitation, transfer to ward, discharge to home etc.

7. The results section is very underwhelming, I'd expect a better reporting of the comparisons between what is deemed clinician performance (also a further explanation of how this was characterized). This again tempers my enthusiasm for the 'superhuman' performance of this algorithm if I don't quite know what humans you are comparing against.

7. PLOS authors have the option to publish the peer review history of their article (what does this mean?). If published, this will include your full peer review and any attached files.

Reviewer #1: No

---

## [Author Response · Author response to Decision Letter 1]

1 Oct 2021

Please see attached file "Answer_Reviewers.pdf"

---

## [Decision Letter · Decision Letter 2]

8 Nov 2021

PONE-D-21-12126R2Superhuman Performance for Sepsis Treatment by Distributional Reinforcement LearningPLOS ONE

Dear Dr. Pasterk,

Thank you for submitting your manuscript to PLOS ONE. After careful consideration, we feel that it has merit but does not fully meet PLOS ONE’s publication criteria as it currently stands. Therefore, we invite you to submit a revised version of the manuscript that addresses the points raised during the review process.

We look forward to receiving your revised manuscript.

Kind regards,

Chi-Hua Chen, Ph.D.

Academic Editor

PLOS ONE

Reviewers' comments:

Reviewer's Responses to Questions

**Comments to the Author**

1. If the authors have adequately addressed your comments raised in a previous round of review and you feel that this manuscript is now acceptable for publication, you may indicate that here to bypass the “Comments to the Author” section, enter your conflict of interest statement in the “Confidential to Editor” section, and submit your "Accept" recommendation.

Reviewer #1: (No Response)

2. Is the manuscript technically sound, and do the data support the conclusions?

Reviewer #1: Partly

3. Has the statistical analysis been performed appropriately and rigorously? 

Reviewer #1: Yes

4. Have the authors made all data underlying the findings in their manuscript fully available?

Reviewer #1: Yes

5. Is the manuscript presented in an intelligible fashion and written in standard English?

Reviewer #1: Yes

6. Review Comments to the Author

Reviewer #1: Thank you for the response to the reviews, the response mostly addresses the key concerns, however there are still areas where further clarification could help.

1. The title is again a misnomer, as I had previously mentioned there are no existing treatments as such for sepsis, there are however supportive therapies. Regardless of what is in the current ML literature, if the authors are discussing a clinical condition you should thoroughly understand the clinical realities. The difference in the term is quite central to the authors aim, since the authors profess ‘superhuman performance’ for ‘treatment’ one would expect to see improved outcomes, reduced mortality, morbidity etc. None of which is described by this current work.

2. “We want to avoid a medical debate about the choice of our models and try not to approximate medical behavior.”, again this can be misleading, since the authors are claiming a contribution to medical literature the authors should be cognizant of the clinical context and condition their claims upon such findings, such assumptions can be consequential for developing the model.

3. The continued reliance on 90-day mortality causes major concerns for whether the objective of this model is about sepsis or about critical illness in general. I urge the authors to consider the standard definition of 28 days so as to ensure that the policy developed is more closely aligned with sepsis outcomes. This is easily available in MIMIC and the authors should be fully able to extract this data without too much work.

4. Please provide a link to your GitHub repo for assessing other assumptions made in your model.

7. PLOS authors have the option to publish the peer review history of their article (what does this mean?). If published, this will include your full peer review and any attached files.

Reviewer #1: No

---

## [Decision Letter · Decision Letter 3]

22 Jul 2022

PONE-D-21-12126R3Superhuman Performance on Sepsis MIMIC-III Data by Distributional Reinforcement LearningPLOS ONE

Dear Dr. Pasterk,

Thank you for submitting your manuscript to PLOS ONE. After careful consideration, we feel that it has merit but does not fully meet PLOS ONE’s publication criteria as it currently stands. Therefore, we invite you to submit a revised version of the manuscript that addresses the points raised during the review process.

We look forward to receiving your revised manuscript.

Kind regards,

Chi-Hua Chen, Ph.D.

Academic Editor

PLOS ONE

Journal Requirements:

Reviewers' comments:

Reviewer's Responses to Questions

**Comments to the Author**

1. If the authors have adequately addressed your comments raised in a previous round of review and you feel that this manuscript is now acceptable for publication, you may indicate that here to bypass the “Comments to the Author” section, enter your conflict of interest statement in the “Confidential to Editor” section, and submit your "Accept" recommendation.

Reviewer #1: All comments have been addressed

2. Is the manuscript technically sound, and do the data support the conclusions?

Reviewer #1: Yes

3. Has the statistical analysis been performed appropriately and rigorously? 

Reviewer #1: Yes

4. Have the authors made all data underlying the findings in their manuscript fully available?

Reviewer #1: Yes

5. Is the manuscript presented in an intelligible fashion and written in standard English?

Reviewer #1: Yes

6. Review Comments to the Author

Reviewer #1: Thank you for your careful review and response of the revisions. Most comments have ben addressed satisfactoriy. Please make mention of the your response #1.10 in the actual manuscript and emphasize this distinction in comparison to those stated works.

7. PLOS authors have the option to publish the peer review history of their article (what does this mean?). If published, this will include your full peer review and any attached files.

Reviewer #1: No

---

## [Decision Letter · Decision Letter 4]

15 Sep 2022

Superhuman Performance on Sepsis MIMIC-III Data by Distributional Reinforcement Learning

PONE-D-21-12126R4

Dear Dr. Pasterk,

We’re pleased to inform you that your manuscript has been judged scientifically suitable for publication and will be formally accepted for publication once it meets all outstanding technical requirements.

Kind regards,

Chi-Hua Chen, Ph.D.

Academic Editor

PLOS ONE

Additional Editor Comments (optional):

Reviewers' comments:

Reviewer's Responses to Questions

**Comments to the Author**

1. If the authors have adequately addressed your comments raised in a previous round of review and you feel that this manuscript is now acceptable for publication, you may indicate that here to bypass the “Comments to the Author” section, enter your conflict of interest statement in the “Confidential to Editor” section, and submit your "Accept" recommendation.

Reviewer #1: All comments have been addressed

2. Is the manuscript technically sound, and do the data support the conclusions?

Reviewer #1: Yes

3. Has the statistical analysis been performed appropriately and rigorously? 

Reviewer #1: Yes

4. Have the authors made all data underlying the findings in their manuscript fully available?

Reviewer #1: Yes

5. Is the manuscript presented in an intelligible fashion and written in standard English?

Reviewer #1: Yes

6. Review Comments to the Author

Reviewer #1: Thank you to the authors for addressing all the comments as suggested. No further revisions are necessary.

7. PLOS authors have the option to publish the peer review history of their article (what does this mean?). If published, this will include your full peer review and any attached files.

Reviewer #1: No

---

## [Editor Report · Acceptance letter]

21 Oct 2022

PONE-D-21-12126R4 

Superhuman Performance on Sepsis MIMIC-III Data by Distributional Reinforcement Learning 

Dear Dr. Pasterk:

I'm pleased to inform you that your manuscript has been deemed suitable for publication in PLOS ONE. Congratulations! Your manuscript is now with our production department. 

Kind regards, 

on behalf of

Professor Chi-Hua Chen 

Academic Editor

PLOS ONE